# Optical Writing and Electro-Optic Imaging of Reversible Space Charges in Semi-Insulating CdTe Diodes

**DOI:** 10.3390/s22041579

**Published:** 2022-02-17

**Authors:** Adriano Cola, Lorenzo Dominici, Antonio Valletta

**Affiliations:** 1Institute for Microelectronics and Microsystems, IMM-CNR, Via Monteroni, 73100 Lecce, Italy; 2Institute of Nanotechnology, NANOTEC-CNR, Via Monteroni, 73100 Lecce, Italy; lorenzo.dominici@nanotec.cnr.it; 3Institute for Microelectronics and Microsystems, IMM-CNR, Via Del Fosso Del Cavaliere, 100, 00133 Rome, Italy; antonio.valletta@artov.imm.cnr.it

**Keywords:** optical doping, polarization, Pockels effect, radiation detectors, CdTe, electric field, deep levels, Schottky contact, semi-insulating

## Abstract

Deep levels control the space charge in electrically compensated semi-insulating materials. They limit the performance of radiation detectors but their interaction with free carriers can be favorably exploited in these devices to manipulate the spatial distribution of the electric field by optical beams. By using semi-insulating CdTe diodes as a case study, our results show that optical doping functionalities are achieved. As such, a highly stable, flux-dependent, reversible and spatially localized space charge is induced by a line-shaped optical beam focused on the cathode contact area. Real-time non-invasive imaging of the electric field is obtained through the Pockels effect. A simple and convenient method to retrieve the two-dimensional electric field components is presented. Numerical simulations involving just one deep level responsible for the electrical compensation confirm the experimental findings and help to identify the underlying mechanism and critical parameters enabling the optical writing functionalities.

## 1. Introduction

Semiconductor devices rely on electric fields in order to provide suitable paths for the free charge carriers. The charge flows are strictly related to the control of the space charge defined by the doping structure design. Whatever the origin and the nature of atomic doping, fixed charges are introduced in the regions of the devices, which define and, to some extent, limit, the flexibility of the device functionalities. A viable route to overcome this limitation is offered by optical doping, defined here as an optical perturbation leading to a spatially localized and stable modification of the charges. In this regard, scanning light beams have been recently used to permanently write monolithic integrated circuits on a two-dimensional semiconductor, via irreversible processes such as direct defect creation [1] or assisted by thermally activated reactions [2]. Permanent optical doping was also realized in the active channel layer of thin films transistors resulting in the enhancement of their parameters [3]. Reversible optical doping has been often shown in monolayers, such as WS_2_ [4] or, more frequently, graphene [5,6,7,8,9].

In this paper, we demonstrate the possibility of stable and reversible optical doping at a bulky level by means of the very well-known and mostly undesired companions of every semiconductor platform, which are the deep levels. Generally speaking, deep levels are detrimental in the world of semiconductor devices, and the fight for their suppression has been ongoing for decades. Deep levels also affect the most-developed technological platform, silicon, when considering that the manufacture of nanometer-sized transistors requires quasi-atomic accuracy. In semiconductor compounds, such as GaAs, InP, CdTe, and GaN, which are of paramount relevance for micro- and opto-electronics, deep levels are inherently present under considerable concentrations that affect, and often limit, the devices’ performances, although deep levels are exploited when aiming towards maximum electrical resistivity, resulting in semi-insulating materials when the energy gap is sufficiently large. It is worth noting that the majority of semiconductor devices lie on semi-insulating substrates. They enable the growth of high-quality epitaxial active layers while ensuring electrical insulation and mechanical support. Semi-insulating materials are not only passive elements such as substrates, but are also largely employed as radiation detectors, where high electric fields are applied to bulky crystals through ohmic or blocking electrical contacts. Conversely, one of their main limitations relies in the so-called radiation-induced polarization [10] effect, which severely deteriorates the detection performance of CdTe, CdZnTe, and other materials when exposed to high X-ray fluxes. Similar detector degradation, called bias-induced polarization [11], also occurs under dark conditions because of the voltage biasing, but on a much longer timescale. Despite being frequently independently investigated, it was quickly recognized that both effects are related to charges trapped [10,12] by deep levels. The present work finds its original motivation precisely in such effects, offering a brand new perspective on the potentialities of high concentrations of deep levels, in terms of optical control and manipulation of the space charge distribution. Specifically, we adopt here semi-insulating CdTe diode-like radiation detectors as a case study. They are described in the next section where we also present our experiments, based on the perturbation of the internal electric field by a focused optical excitation, which allowed us to durably modify the electrostatic charge distribution in a highly non-uniform fashion in space. Remarkably, the optical excitation occurs through the top planar semitransparent cathode and the space charge can be ‘written’ arbitrarily below it, close to the anode. The electric field is observed by means of an advanced electro-optic imaging system through the Pockels effect, which provides a powerful tool to precisely map the real-time evolutions along the transverse plane with good 2D spatial resolution. In the Results and Discussion section, electric field transients and spatial profiles are analyzed and the effect of external parameters such as the incident optical irradiance and temperature is discussed. The electric field maps were calculated based on an original reconstruction procedure.

It will be also shown that the observed experimental features are consistent with numerical simulations, including the stable optical doping effect, thus paving the way to fully exploit the potentialities offered by systems controlled by deep levels.

In Appendix B, we provide a quick excursus through electrical compensation, deep levels, and blocking contacts in order to account for the out-of-equilibrium properties of CdTe detectors, with specific reference to the behavior of the biased detector under dark conditions. Appendix B also constitutes the framework for understanding how optical doping can be efficiently realized in systems controlled by a single deep level.

## 2. Materials and Methods

The experimental setup is sketched in Figure 1. The sample is a 4 × 4 × 1 mm^3^ slab CdTe diode-like X-ray detector from Acrorad Co., Ltd. (Okinawa, Japan), equipped with ohmic Pt and Schottky In planar contacts on the upper and lower square surface, respectively. The detector is a CdTe:Cl semi-insulating crystal, slightly p-type, and 111 oriented, and the electric field is applied vertically (*y*-axis) between the anode (In) and cathode (Pt) along the direction perpendicular to the 111 plane of the crystal. Two optical beams are impinging on the sample, the probe and the excitation beam. The wavelength of the first beam is 980 nm, and is thus not absorbed by the sample (*E_g_* = 1.43 eV, corresponding to λ_g_ = 840 nm), in order to probe the internal electric field by means of the Pockels effect in a transmission configuration. This beam, linearly polarized at 45° with respect to the *y*-direction, impinges orthogonally on the lateral surface of the detector and experiences the electric field birefringence when crossing the sample. The effect is detected due to a second polarizer (analyzer) oriented at −45°, placed behind the sample. The images *P*(*x*,*y*) of the transmitted intensity are recorded by a high sensitivity monochrome camera equipped with a 4.5× zoom lens, suitable to image the 1 mm detector thickness. The intensity transmitted throughout cross-polarizers is normally zero, but changes in the presence of internal birefringence, according to a well-known modulational formula [13], and the scheme is able to convert any phase difference (here between the *xy* linear polarizations) into an intensity signal. When assuming the electric field uniform along the direction of the beam propagation (*z*), the *P*(*x*,*y*) images are hence related to the *E*(*x*,*y*) electric field distribution according to [14]:(1)$$P\left(x,y\right)=P_{para}\left(x,y\right)\sin^{2}\left[\frac{\pi }{2}\frac{E\left(x,y\right)}{E_{0}}\right]$$

Here $P_{para}\left(x,y\right)$ is the maximum transmitted intensity of the probe, taken in the parallel configuration of the two polarizers (both at 45° and in the absence of any induced birefringence, i.e., with no applied voltage); such a reference image is presented in the top-right of Figure 1. *E*_0_ is a constant given by:(2)$$E_{0}=\frac{\lambda }{\sqrt{3}n_{0}^{3}r_{41}L}$$
where *n*_0_ is the field free refractive index in CdTe, *r*_41_ its linear electro-optic coefficient, *L* = 4 mm is the physical path length through the crystal, i.e., the detector z-depth, and *λ* the wavelength of the incident probe light. Assuming *n*_0_ = 2.8 and *r*_41_ = 5.5 × 10^12^ mV^−1^ [15] results in *E*_0_ = 11.7 kV/cm. The meaning of *E*_0_ is that it represents the local electric field value that allows a complete polarization inversion of the crossing probe beam, hence the Pockels image bears maxima (equal to *P_para_*) and minima (zero) intensity values at the transverse *xy* positions where the internal electric field is equal to odd and even multiples of *E*_0_, respectively. When in the presence of local electric field values greater than *E*_0_, multiple fringes appear in the Pockels *P*(*x*,*y*) images and an unwrapped-like reconstruction is needed to achieve the *E*(*x*,*y*) profile.

The other optical beam used in our setup is the excitation beam, which is line-focused on the cathode side along the *z*-direction (see Figure 1) upon using a cylindrical lens, thus preserving the electric field uniformity along such direction. The beam is about 150 μm wide and more than 4 mm long, thus impinging along the whole detector length. The optical perturbing beam comes from a supercontinuum laser, which is a laser exhibiting a broad white spectrum because of non-linear processes acting upon a pump beam into a photonic optical fiber. Its spectrum was filtered by a long-pass filter at 780 nm cut-on wavelength. This enables both a good transmission through the top semi-transparent planar Pt contact, realized via electroless plating, and a high absorption and carrier generation within a few microns of the CdTe’s depth (see Appendix A).

The supercontinuum laser is pulsed with a 1–2 ns width and 24 kHz repetition rate. Here, we will not investigate the electro-optical response of the CdTe on the ns timescale duration of the pulses, which is that of the free carrier dynamics, nor the microsecond timescale associated with the multiple pulses. As we are interested in the long terms effect of the ‘integral’ irradiation, the pulsed nature of the beam is not relevant.

The general procedure of the experiment is the following. After recording the image corresponding to the maximum transmission, i.e., with the two polarizers set parallel at 45° and no applied voltage, a sequence starts (see Figure 2) with the crossed polarizers, where the detector is progressively biased up to 600 V in steps of 100 V. As the crystal is slightly p-type, in order to reverse bias the diode, the positive voltage is applied on the Indium hole-blocking contact, hereafter called the anode. After 5 min under dark conditions at 600 V, the excitation beam is switched on and kept shining for 5 min, then switched off for 15 min, and then the detector is biased back to 0 V in steps of 100 V. During the whole sequence, the electro-optic images are recorded every 2 s, at a fixed exposure time ranging between 50 ms and 150 ms depending on the specific experiment.

In different experiments, the incident irradiance was varied by 3–4 orders of magnitude by means of neutral density filters, and some measurements were also carried out at different temperatures, between 20 and 50 °C.

## 3. Results and Discussion

### 3.1. Main Effects of Optical Irradiation

We report in Figure 3 four Pockels images all under 600 V but at specific times of the sequence (marked with red dots in Figure 2), corresponding to these situations: just after the 600 V bias is applied (panel a); just before the optical perturbation (b); at the end of the irradiation interval (c), which lasts 5 min; and 15 min after the end of the irradiation (d). The complete sequence of this experiment can be seen as a movie in the Appendix A where all images have been normalized by $P_{para}\left(x,y\right)$.

It can be immediately noted that the two first panels (Figure 3a,b) look similar to each other, with a bright region extending vertically from the anode side and uniform along the horizontal direction. The brighter region close to the bottom anode represents the presence of the electric field in that region, imaged through the electro-optic effect. By comparison, after the application of the focused optical beam, the map shows a substantially different situation (Figure 3c). The Pockels map becomes highly perturbed, showing a central dark area where before it was light, and most interestingly, a number of fringes close to the bottom anode and at central x. The fringes, according to Equations (1) and (2), now indicate the presence of local electric fields larger than in the first two maps, and increasing up to different multiples of *E*_0_. Substantially, the main consequence of the optical irradiation, which occurs on the cathode side (and whose section is represented as a red segment in Figure 3b) is a huge increase in the electric field close to the anode, maximum at the transverse position of irradiation x_irr_ and, as expected, laterally symmetric with respect to this axis. Furthermore, a region of negligible electric field has formed almost circularly on the cathode side.

For the same instants, the vertical crosscuts of the maps were used to retrieve the central electric field profiles that we report in Figure 3e–h. The two profiles corresponding to the pre-irradiation times (Figure 3e,f) are basically linear and their slopes tend to slightly increase with time, consistently with the bias-induced polarization due to the hole emission from the deep level (see Equation (A5)).

Looking at the field profiles at the end (Figure 3g) and after the irradiation (Figure 3h), we note again that the effect of irradiation is to shrink the field towards the anode, where it becomes as large as 75 kV/cm, whereas the field becomes negligible across most of the detector thickness, except for a weak build-up close to the cathode. However, the most striking result is that the strong perturbation of the electric field persists almost unaltered after the irradiation, in every point of the detector, as can be seen by comparing Figure 3c,g with Figure 3d,h. The process is reversible: a voltage reset, by erasing the space charge, quickly restores the initial conditions and a successive irradiation experiment under bias produces once again the same results.

The shrinking effect of the electric field towards the anode has been already reported in the case of uniform optical irradiation [16,17] and it is coherent with the same charge state modification of the deep level occurring under dark, i.e., with the increase in its negative space charge. Such an increase is due to the great number of photo-generated electrons initially flowing from the irradiated cathode region. Whereas the hole emission is the dominant process under dark conditions, it is under irradiation that electron capture plays a major role, and its rate is dependent on the electron concentration. In other words, the space charge evolution is still described by Equation (A4) for the same deep level, but the rate is given by $C_{n}n$ instead of $e_{p}$. Hence, the deep level is able to communicate with both the valence and conduction band like a pure recombination center, but still retains the two charge states typical of traps.

Just after the light is switched off, the electric field appears to be only slightly affected. Then, it remains almost unaltered for 15 min after the end of irradiation (see maps in Figure 3c,d and profiles in Figure 3g,h), for almost any spatial point, which indicates the stability of the space charge profile set at the end of the irradiation. Actually, after switching off high optical fluences (integrated irradiance during the time of exposure), residual electric field variations at the anode smaller than a 0.5% were measured in experiments lasting one hour or more. This is consistent with the acceleration in the space charge increase provided by the electron capture: when irradiation stops at *t_stop_*, the space charge evolves with the slow rate $e_{p}$ following Equation (A5), starting from the initial condition $N_{da\left(t=t_{stop}\right)}^{-}$. This keeps further changes in the space charge limited to $\left(N_{da\left(t=t_{stop}\right)}^{-}-N_{da}\right)$, which can be possibly very small after exposure to large fluences.

When the diode is biased under dark, hole emission is the dominant, temperature-activated, process [11]. This is further confirmed by measurements carried out at different temperatures, from 20 to 50 °C (see Appendix A), where, after each voltage step, the electric field transients are faster at higher temperatures. By comparison, when light is shined on the device, the time constant of the electric field is very fast and does not undergo appreciably change at higher temperatures. This is ascribed to the temperature-independent electron capture process, which prevails over the thermal hole emission. After irradiation, the maps of the electric field remain practically frozen to the last instant of the optical irradiation for all temperatures.

Our results indicate that, under both dark conditions and optical activation, the space charge tends towards the same steady state (Q_ss_), which is set by the applied voltage, eventually corresponding to the full ionization of the deep acceptor $\left(N_{da\;}^{-}=N_{da}\right)$, and so does the electric field distribution. Under dark conditions, the rate of the process is very slow, increasing with temperature. Under an optical beam, the rate raises substantially, depending on the irradiance level. Indeed, the transients of the electric field close to the anode in Figure 4 show both a larger step and speed with increasing levels of irradiance. Figure 4 also shows that, as a consequence of the increased negative space charge, the electric field levels reached at the anode increase tending to a saturation. Saturation behavior is also observed for the associated space charge under high irradiance levels, as shown in the inset of Figure 4. This is a signature that, under reaching such a circumstance, the deep level becomes completely electron filled in the space charge region. Again, cathode irradiation looks like a kinetic factor that accelerates the process of achieving the stationary condition. After switching the light off, the space charge remains there under dark conditions, while its residual build-up process is so slow that measures prolonged over hours only showed very small variations.

A small and fast bump is noticed in Figure 4 at the end of the optical irradiations, which grows with irradiance. This is related to the free carriers present on the cathode side being quickly swept out, thus leaving the electric field being only determined by the fixed charges. This point will be further addressed in the Numerical Simulation Subsection.

In summary, what is usually called bias- or radiation-induced polarization, represents in both cases the evolution towards the full ionization of the deep level responsible for the electrical compensation, resulting in an electric field strongly confined under the anode. The detectors work the best when the electric field extends uniformly throughout its whole thickness. However, because voltage is applied, either under dark conditions and under irradiation, they work in a non-stable situation, which degrades during operation only by gradually shifting it towards the stationary point. In terms of radiation-induced polarization, it should be remarked that some difference might be expected when dealing with X-rays, which are characterized by penetration depths that are much longer than the optical photons considered here.

From a different perspective, these results highlight the potentialities offered by controlling the space charge upon application of an optical excitation. The biased detector works as a reservoir of space charge, which can be activated and drawn locally close to the anode side in correspondence to the irradiated position on the opposite cathode. The persistence of the induced charge when irradiation is switched off enables an optical memory functionality. Additionally, with voltage kept applied, successive optical irradiations will add further local space charge, until the maximum level Q_ss_ is achieved. Such a property can be exploited, for example, in dose-meter applications because the total space charge depends on the integrated optical flux. Conveniently, as a read-out tool, the Pockels effect allows us to directly monitor the local electric field, and hence the local space charge, in any instance, without affecting it.

When switching off the voltage bias, independently of temperature or voltage, the net space charge and the electric field is nulled everywhere in the device (except built-in values very close to the electrodes), resulting in completely erasing the memory of previous optical irradiations. Notably, multiple optical irradiations at different x coordinates can be exploited to control the spatial distribution along the *x*-direction (see Appendix A as an example of two successive irradiations at different points). If irradiation was carried out using focused spots instead of lines, the space charge could be arbitrarily written with resolution in the x-z plane instead of along the *x*-direction only, while still keeping the charge integration functionality.

### 3.2. Electric Field and Space Charge Maps

The Pockels effect has been extensively used to evaluate the electric field profiles between planar electrodes (i.e., field only along the *y*-direction, as in our case before the applied optical perturbation) in CdTe-based radiation detectors [16,18]. In a recent paper by Dědič [19], a non-homogeneous x-y electric field was analyzed and imaged in 111 CdTe crystals, as in our case. A strip electrode was present along the *z*-direction, rather than our line-shaped optical perturbation, in order to introduce the non-uniformity. As noticed by Dědič [19], the vertical component *E_y_*(*x*,*y*) only is obtained from Equations (1) and (2) when using the diagonal configuration, i.e., with the probe beam linearly polarized at 45° with respect to the vertical direction y. Dědič also calculated, for the 111 CdTe crystal, the numerical relation between the angle of the first polarizer (the analyzer still being perpendicular to it) and the weights of the *E_x_* and *E_y_* electric field components that are contributing to build up the electro-optic image intensity.

Here, we follow an alternative single measurement approach to obtain the missing *E_x_*(*x*,*y*) component, starting from the *E_y_*(*x*,*y*) one, which is that imaged in the configuration with the first polarizer set at 45°. We rely on the property of the electric field being a conservative field, meaning that it can be described as the derivative of a scalar potential. Upon integrating the *E_y_* component along the *y*-direction for each vertical profile (i.e., at any *x*), we can hence build back the full potential map *V*(*x*,*y*). We have used for the initial condition the approximation that the field is null at the top cathode electrode. It is then possible to apply the derivative of the potential along the *x*-direction to obtain the missing component, *E_x_*(*x*,*y*) = −*dV*(*x*,*y*)/*dx*.

To firstly achieve the *E_y_*(*x*,*y*) component, we initially normalized the sequence of cross-polarized images $P\left(x,y\right)/P_{para}\left(x,y\right)$ by using the parallel configuration image. At this point, it is needed to reverse apply the modulation formula, but also to unwrap the E values across multiple fringes. This is not always straightforward. Furthermore, due to experimental limitations, the visibility of the fringes can be notably reduced in the region of their maximum density (due, for example, to resolution factors). This would strongly affect the field retrieval. Thus, we decided instead to fit the normalized experimental image along the vertical profiles. Upon using a two-factor function directly for the electric field, on top of which we applied the EO modulation formula, we aimed for the procedure to match the experimental target image. One example of a normalized image (corresponding to the situation at the end of the optical excitation, i.e., to data in Figure 3c,g) and its fit are presented in Figure 5a,b, respectively. We note the importance of the experimental minima and maxima in the Pockels image, where they are unambiguously associated with multiple values of *E*_0_. Figure 5c reports the distribution of the electric field intensity as a false color map, showing its higher concentration close to the bottom anode. The direction of the electric field is represented by the superimposed streamlines, which were retrieved using the condition on the conservative nature of the field as described above (see also Appendix A for a map of the two components). Finally, we applied the divergence to the vector electric field in order to achieve a representation map for the excess spatial charge, as shown in Figure 5d. Interestingly here, the maximum localization of the space charge happens to be at some finite distance from the bottom anode electrode. Its transverse profile (horizontal cut across the maximum) is well reproduced by a Gaussian curve.

The calibration of the electric field units was performed by matching the field integral condition $\int_{0}^{L}E_{y}\left(x,y\right)dy=V$, where *V* = 600 V is the applied voltage, whose uniformity is well observed at the lateral boundaries. The retrieved *E*_0_ is 10–13.7 kV/cm, in very good agreement with the value expected from Equation (2). To compute the spatial charge map *N*(*x*,*y*) we used the formula of the field divergence *N*(*x*,*y*) = *ε*/*q* × (*dE_y_*/*dy* + *dE_x_*/*dx*), with *ε* = (10.3 × 8.85) 10^−14^ F cm^−1^.

### 3.3. Numerical Simulations

Two-dimensional numerical simulations was performed using the semiconductor device simulator “Sentaurus”, part of the Technology CAD software package provided by Synopsys, Inc. [20]. By starting from a reliable model for the CdTe semi-insulating diode in the dark, a uniform optical irradiation through a 150 μm wide window on the cathode was then implemented.

Without attempting overly onerous best-fitting procedures, the simulations allowed the inference of meaningful ranges for some critical parameters within a two-level model (shallow donor and deep acceptor) that was able to reproduce not only the main experimental features observed in the time sequences, but also in the electric field profiles and, reasonably, in the maps. In particular, this was obtained with concentration differences $N_{da}-N_{sd}\sim \;\left(3-4\right)\cdot 10^{13}\;{\mathrm{cm}}^{-3}$, with *N_sd_* being a few 10^13^ cm^−3^. It was found to be crucial to use comparable values of deep level capture cross-sections $\sigma_{p},\;\sigma_{n}$ around $10^{-18}–10^{-19}\;{\mathrm{cm}}^{2}$. As capture cross sections directly affect the rates of space charge variations (see Equation (A2)), they are key parameters in our experiments dominated by transient effects.

We remark that, in addition to the present experimental results, different analyses [21,22] of similar material showed that the deep level was able to communicate with both valence and conduction bands, which implies comparable capture cross-sections.

According to previous experiments on similar materials, the energy of the deep acceptor was fixed to *E_da_* = 0.725 eV from the valence band [22] and the electron Schottky barriers for indium and the platinum contacts at 0.5 and 0.8 eV [21], respectively. We note that the semi-insulating property within our two-level model is mainly ensured by the proper combination of the *N_sd_*, *N_da_*, and *E_da_* parameters [21]. With the Fermi level being around mid-gap, the considered Schottky barrier values account for the hole-blocking nature of the Indium contact and the slightly hole-injecting character of the Pt contact [23]. As previously shown [21], such a combination of parameters accounts for the completely different electric field profiles experimentally observed among In/CdTe/Pt and Pt/CdTe/Pt detectors.

In order to proper simulate the optical irradiation, the spectral distribution of the incident radiation and the CdTe absorption coefficient [24] are accounted for by the simulation. A scaling factor was introduced in the simulated optical irradiance to account for the contact transparency.

The whole time sequence, as reported in Figure 2, was simulated for a proper comparison with the experimental results.

The results of numerical simulations reported in Figure 6, which show the time evolution of the electric field at the anode in correspondence with the center of irradiation (x_irr_) under similar conditions to those corresponding to Figure 4, show a good agreement with the experimental results. In particular, this concerns the voltage steps under dark conditions and the slow transience observed at 600 V, then the increase in the electric field under different levels of optical irradiance, both in terms of time constant and growth level. As in Figure 4, the inset of Figure 6 reports the space charge computed at the anode at the end of the irradiations, which confirms the agreement with the experiments. Importantly, the stability of the electric field after the irradiations is confirmed by the simulations. Electric field profiles at different instants are also comparable with the experiments (see Appendix A). In particular, a secondary feature is also confirmed, which consists in the weak field build-up close the cathode, which is especially noticed under irradiation. The effect, more pronounced in the experiments than in the simulations, is associated with and sensitive to the slight upward band bending expected for the platinum contact [23]. Finally, we report in Figure 7 the maps of the electric field and space charge simulated under the same experimental conditions of Figure 5. In particular, Figure 7a refers to the module of the electric field at the end of optical irradiation, which thus can be directly compared with Figure 5c. Analogously, in Figure 7b, the map of the simulated space charge can be compared with the experimental one in Figure 5d.

When comparing the two maps, we should take into account that the map of the space charge in Figure 5d is subject to some limitations due to both the heuristic fitting functions and because these are applied to experimental images, whose fringe visibility is resolution limited in the region of the most intense field.

However, it can be seen that the agreement is also favorable in 2D space. Remarkably, the simulations confirm the localization of the field close to the anode and its peak density value, in addition to the localization of the space charge with a maximum at a few tens of µm internal position. In Figure 7c, the horizontal crosscut of the space charge maps is plotted; for the sake of comparison, both simulated and experimental profiles are reported. One main difference can be noted in that the experimental profile is that of a Gaussian shape, whereas the numerical one is slightly flattened in the central zone. We ascribe this to a non-perfect tuning of the parameters, which in this case are seemingly describing a central saturation of the space charge. Simulations at a lower irradiance show a Gaussian transverse space charge profile. On another note, it should be mentioned that the non-uniform space charge vertical profile (i.e., an electric field not linearly varying in space) in the space charge region is not predicted in the approximate full-analytical model expressed by Equation (A7). This is arguably related to the boundary conditions at the anode affecting the charge terms in the Poisson equation.

Many other interesting features emerge from the numerical simulations but their analysis is behind the scope of this paper. Here, we just point out the role of carrier diffusion.

In previous papers, spatially uniform pulsed [17] and constant [16] irradiations were performed on the cathode of CdTe diode-like detectors, and it was inferred that electron diffusion was the main transport mechanism close to the cathode. In the present work, the maps in Figure 5c and Figure 7a show a circular region around the optical irradiation window where the electric field becomes negligible and diffusion prevails. However, during the optical irradiation, a great number of electron hole pairs are created and simulations show that the hole concentrations (see Appendix A) and their associated diffusion currents (Appendix A) compete with the electron ones in such a region. Moreover, the net concentration of free carriers is well balanced by the fixed charges provided by the deep level, except when very close to the cathode, where we have already noticed a weak field build-up, indicative of a small positive charge (see the inset of Appendix A). This is consistent with the large quasi-neutral region extending around the irradiation window. When the optical irradiation is switched off, the excess of free carriers quickly disappears, either by free carrier recombination or by the deep level trapping, still maintaining the charge neutrality in the same region as during the optical irradiation. As the electric field is mainly determined by the negative space charge in the depletion region under the anode, no appreciable changes occur at the anode when switching off the irradiations. However, the simulations show that complex carrier dynamics is occurring, especially just after the light switching. Hence, it is not surprising that, in contrast to the simulations, appreciable variations are detected in the anode electric field at the irradiation switch off. Moreover, non-idealities, as thin interfacial layers, which are known to be present at the contacts [11,25], could, for example, play a role by increasing the surface recombination velocity and thus distorting the electric field distribution.

## 4. Conclusions

We have shown that optical doping is feasible across the planar electrode surface of CdTe diode-like detectors, offering stable, additive, and erasable space charge in regions locally exposed to optical beams. In particular, the space charge that appears to be localized just under the anode, in correspondence with the irradiated cathode, originates from the ionization of the deep level responsible for the electrical compensation. The good agreement between the experiments and simulations highlights the strength of our simple two-level model and the interpretation based on the deep level communications with both extended bands. Furthermore, it indicates the possibility of designing new devices, which may be eventually miniaturized, or of exhibiting a larger dynamic range in order to exploit the effect in specific applications, such as dose meters, imaging detectors, optical memories, or elaborations. We remark that using the Pockels effect as a space charge probing tool provides a great advantage not only because of its spatial resolution, but also because it allows the reading of the space charge written by the optical irradiation, without perturbing it. In this regard, we also presented a simple method to accurately reconstruct the vector map of the two-dimensional internal electric field based on the conservative nature of the electric field. As a concluding remark, it is possible to state that, although high concentrations of deep levels are deleterious in semi-insulating radiation detectors, they can provide new perspectives in optical doping.

## Figures and Tables

**Figure 1 sensors-22-01579-f001:**
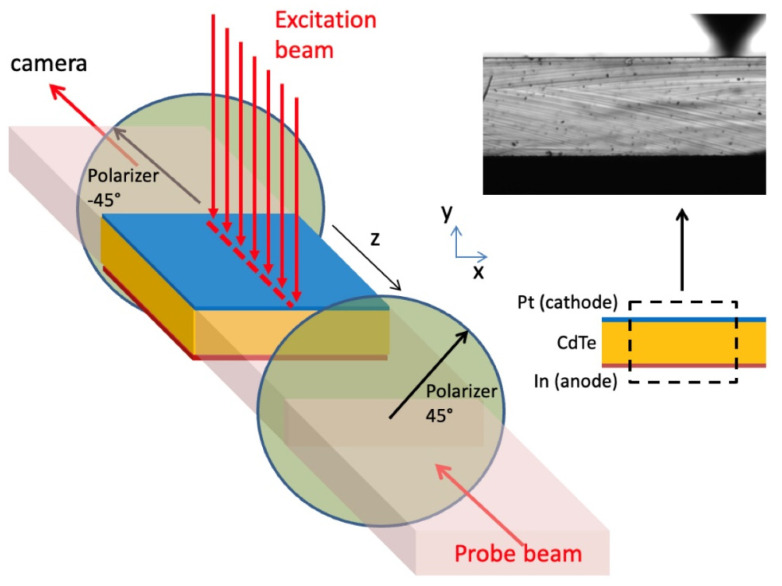
A sketch of the experiment: the line-focused optical excitation beam is incident on the cathode side of the CdTe diode-like detector (left), whereas the 980 nm probe beam laterally crosses the detector in a cross-polarizer configuration. On the right, a picture of the detector section and above it an image of the transmitted intensity with both polarizers aligned and no voltage applied (*P_para_*).

**Figure 2 sensors-22-01579-f002:**
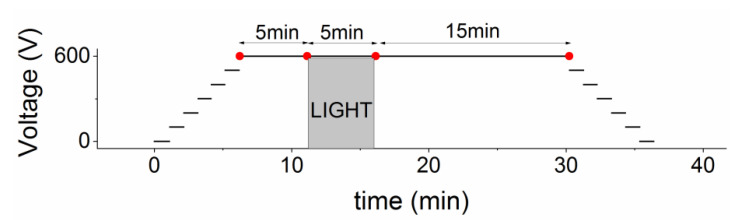
The voltage and light excitation sequence of the experiment. Results corresponding to red points are shown in Figure 3.

**Figure 3 sensors-22-01579-f003:**
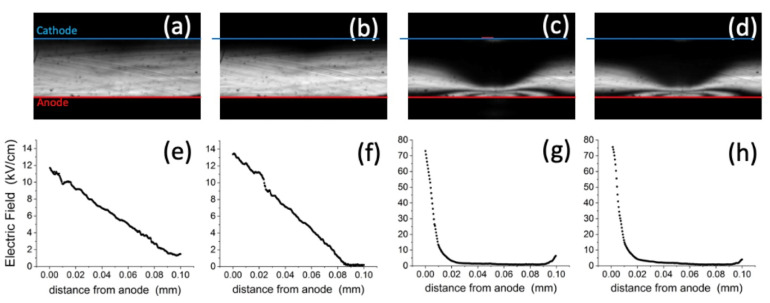
Pockels image at specific times (marked with red points in Figure 2): after the bias ramp reached 600 V (**a**), just before the optical perturbation (**b**), at the end of the irradiation interval, which lasts 5 min (**c**), and 15 min after the end of the irradiation (**d**). For the same times, (**e**–**h**) report the central profiles of the vertical electric field at the transverse coordinate x_irr_ of irradiation (center of red bar in panel c, representing the width of the optical beam). In order to improve the quality of the line profile calculation, the values of 20 columns centered around x_irr_ were averaged. For such an x range, about 80 µm, the electric field is fundamentally vertical. The temperature was 40 °C and a neutral filter of optical density 0.2 was used.

**Figure 4 sensors-22-01579-f004:**
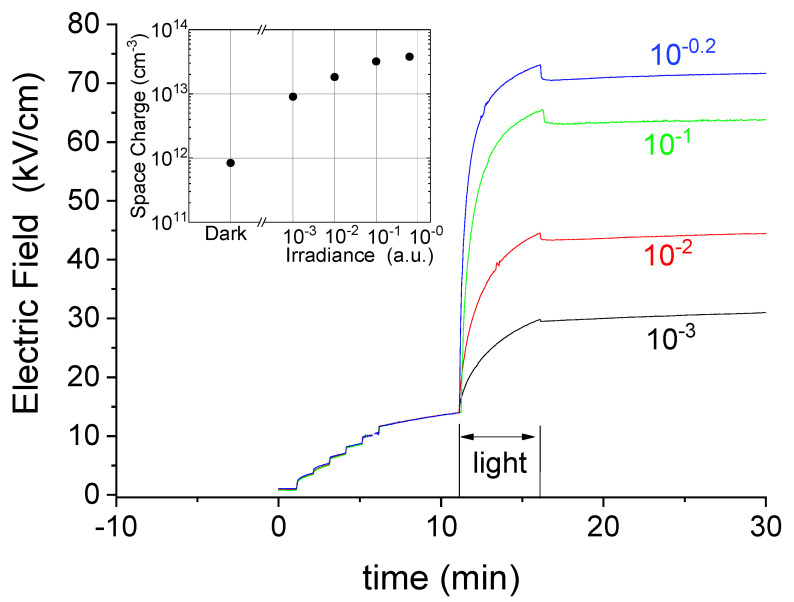
Electric field transients at the anode during the experiments under different optical irradiance. In the inset, the space charge under dark conditions (just before light) and under optical excitation, just before switching it off. The temperature was 40 °C. For each transient, the neutral filter optical density is reported.

**Figure 5 sensors-22-01579-f005:**
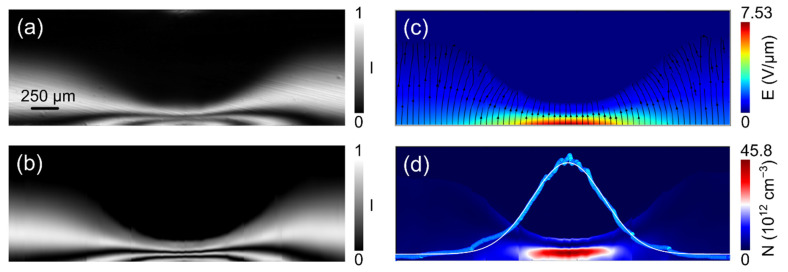
Electro-optic imaging and field retrieval: (**a**) the experimental normalized image, (**b**) the image obtained in the fit procedure, (**c**) the associated electric field map with streamlines indicating its local direction, and (**d**) associated space charge map with its profile (blue dots) along a horizontal crosscut passing through the maximum, at around 70 µm from the bottom anode. The white solid line is a Gaussian fit of the retrieved charge profile.

**Figure 6 sensors-22-01579-f006:**
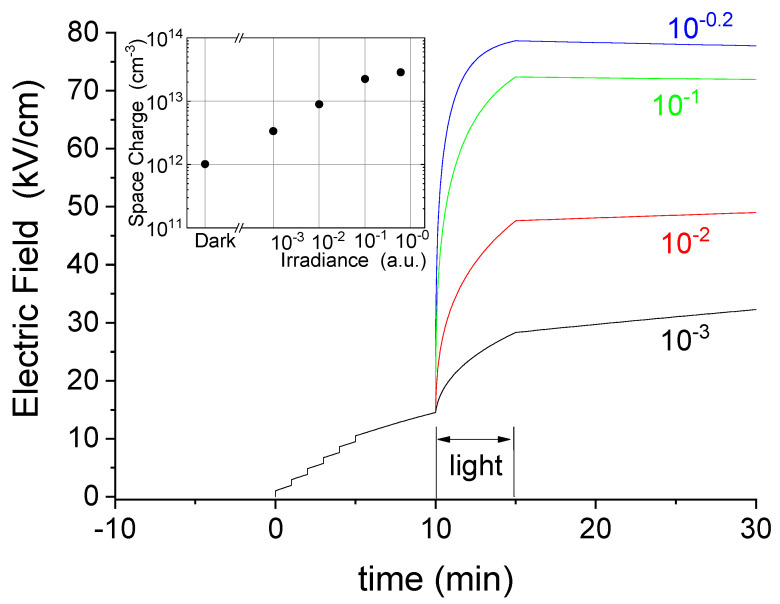
Simulated electric field transients at the anode under different optical irradiances. In the inset, the space charge computed at the anode under dark conditions (just before light) and under optical excitation, just before switching it off. The temperature was 40 °C. Results can be directly compared with the experimental results in Figure 4.

**Figure 7 sensors-22-01579-f007:**
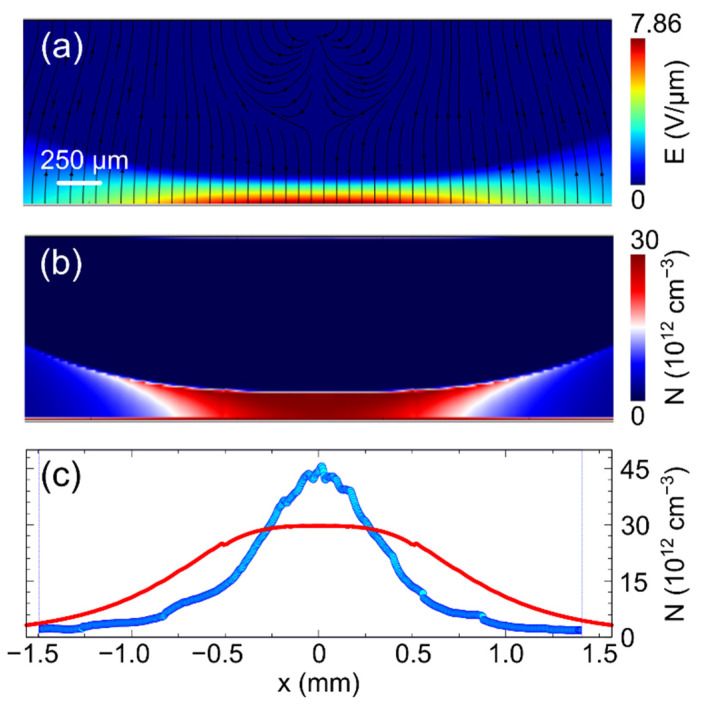
Numerical simulations corresponding to the experimental conditions in Figure 5: (**a**) electric field map with streamlines indicating its local direction, (**b**) space charge map, (**c**) space charge horizontal crosscut (passing by the maximum), red solid line, compared to the experimental horizontal profile, blue dots.

## Data Availability

The data presented in this study are available on request from the corresponding author.

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
