# Peer review of "Optical Writing and Electro-Optic Imaging of Reversible Space Charges in Semi-Insulating CdTe Diodes"

_sensors, 2022, doi:10.3390/s22041579_

Round 1

Reviewer 1 Report

I only have some minor (optional) suggestions for the authors:

Lines 129-133: please clarify the type of laser (meaning of “supercontinuum laser”?) and the details of the 780nm “cut on” filter? Is this a long-pass filter that transmits all wavelengths > 780nm?

L218: please clarify the meaning of “This is consistent with the acceleration provided by the electron capture”

Can the authors give an upper time limit on the duration of the space charge affect after the probe light is switched off. Does it persist far beyond 15 minutes? How does temperature affect this relaxation time constant?

Reviewer 2 Report

The authors present a study of optically induced polarization of semi-insulating CdTe. The paper is of interest for the community. The main novelty is a new approach to calculate 2D electric field distribution from Pockels effect measurements.

I have just a few minor comments

  1. An assumed band scheme of the structure would be useful in the paper
  2. Discussion on lines 428-438 is not clear to me. An increase of electric field observed close to the cathode should be caused by a positive charge. Should not diffusion of holes be considered as the main mechanism of the effect?

Reviewer 3 Report

The paper is interesting and can be published as it is now.

Reviewer 4 Report

Authors report on the investigation of polarization phenomena in CdTe radiation detector. Experiments based on the Pockels effect measurements are properly combined with theoretical simulations. Though the investigated effects are well known, used techniques of their investigation are original. The results are also very well presented in supplementary materials and video. All ideas are well substantiated and consistently discussed. The manuscript should be accepted for the publication in Sensors. I have only minor comments listed below.

1. I agree with the evaluation of the electric field and the space charge from the Pockels data in the depleted layer near the anode as plotted in Fig. 3. However, the profiles near cathode look suspiciously. The illumination of cathode results in a strong supply of electrons flowing (drifting or diffusing) in the inactive region. The excessive negative charge in the inactive layer should result in further decrease of the electric field and not increase, as it is plotted in Fig. 3g. I am pretty sure that the electric field changes the sign near the cathode after the illumination. Authors should check the validity of their results. It is apparent in Eq. (1) that the Pockels effects gives the magnitude of the electric field only. The sign could not be derived directly. 

2. Based on arguments in item 1, it is not clear, how the simulations shown in the inset in Fig. S4 could confirm the experiment. If authors insist on their interpretation, they should add to the supplementary also the zoom of electron and hole densities and drift and diffusion current components near the cathode.

3. Typing errors
Line 118: The symbol of wavelength is missing.
Lines 227-228: '... does not undergoes ...'
Line 360: 'outthat'
Line 555: The symbol of permittivity is missing.
In addition, I noticed a few other minor grammar errors.
